# Peripheral Neuropathies Derived from COVID-19: New Perspectives for Treatment

**DOI:** 10.3390/biomedicines10051051

**Published:** 2022-05-02

**Authors:** Alfredo Córdova-Martínez, Alberto Caballero-García, Daniel Pérez-Valdecantos, Enrique Roche, David César Noriega-González

**Affiliations:** 1Department of Biochemistry, Molecular Biology and Physiology, Faculty of Health Sciences, GIR Physical Exercise and Aging, University of Valladolid, Campus Duques de Soria, 42004 Soria, Spain; danielperezvaldecantos@gmail.com; 2Department of Anatomy and Radiology, Faculty of Health Sciences, GIR Physical Exercise and Aging, University of Valladolid, Campus Duques de Soria, 42004 Soria, Spain; alberto.caballero@uva.es; 3Department of Applied Biology-Nutrition, Institute of Bioengineering, University Miguel Hernández, 03202 Elche, Spain; eroche@umh.es; 4Alicante Institute for Health and Biomedical Research (ISABIAL), 03010 Alicante, Spain; 5CIBER Fisiopatología de la Obesidad y Nutrición (CIBEROBN), Instituto de Salud Carlos III (ISCIII), 28029 Madrid, Spain; 6Department of Surgery, Ophthalmology, Otorhinolaryngology and Physiotherapy, Faculty of Medicine, Hospital Clínico Universitario de Valladolid, 47005 Valladolid, Spain; davidcesar.noriega@uva.es

**Keywords:** α-lipoic acid, COVID-19, nutraceuticals, peripheral neuropathy, vitamins

## Abstract

Peripheral neuropathies constitute a group of disorders affecting the peripheral nervous system. Neuropathies have multiple causes such as infections (i.e., COVID-19), diabetes, and nutritional (low vitamin levels), among others. Many micronutrients, such as vitamins (A, C, D, E, B6, B12, and folate), certain minerals (Fe, Mg, Zn, Se, and Cu), and ω-3 fatty acids have immunomodulatory effects. Therefore, they may play an instrumental role in the treatment of COVID-19 infection. However, many COVID-19 patients can undergo neuropathy. In this context, there is a wealth of information on a variety of first-, second-, and third-line treatment options. This review focuses on the application of nutraceutical strategies in order to improve the symptomatology of neuropathy and neuropathic pain in patients that suffered from COVID-19. Our aim is to provide an alternative vision to traditional medical-pharmacological treatment through nutraceuticals.

## 1. Introduction

Peripheral neuropathy (PN) is a group of nerve disorders affecting the peripheral nervous system (PNS). Neuropathy has several multiple causes including infections, diabetes, and nutritional (low vitamin levels), among others [1,2]. Undoubtedly, following a balanced nutrition is key in the prevention of infections that can cause neuropathy. Recently, Damayanthi et al. [3] highlighted in their review the importance of macro- and micro-nutrients in the nutritional status for COVID-19 management, as well as the importance of optimal protein intake for adequate antibody production.

During an acute inflammatory process following infection, catabolism increases, leading to a rise in muscle protein degradation. C-reactive protein (CRP) and ferritin (both acute-phase reactant proteins) and pro-inflammatory cytokines (tumor necrosis factor-α (TNF-α) and interleukins (ILs)) limit requirements for albumin and muscle protein synthesis [4]. Many of the micronutrients present in food, such as vitamins (A, C, D, E, B6, B12, and folate), certain minerals (Fe, Mg, Zn, Se, and Cu), and ω-3 fatty acids have immunomodulatory effects, thereby being candidates in the treatment of COVID-19 [5]. In addition, unfavorable nutritional status during the development of infection, such as malnutrition or obesity and overweight, can worsen disease progression.

There is a lot of information regarding neurological presentations of PN as well as clinical outcomes, together with a variety of first-, second-, and third-line treatment options. The analysis of the diagnostic and clinical aspects of neuropathies is not the subject of this review, nor even the treatment of neuropathies, as there are already enough publications that deal with this subject. This review is focused on the application of nutraceutical strategies in order to improve neuropathy and neuropathic pain, mainly in COVID-19 patients. The review aims to give an alternative vision to traditional medical-pharmacological treatments, emphasizing the role that nutraceuticals can play in controlling pain and helping to restore nerve function. This will undoubtedly lead to an improvement in patients’ quality of life.

## 2. Peripheral Neuropathy

The somatic nervous system (SNS), as part of the PNS, works as the motor controller of the organism. SNS receives information from the dorsal (posterior) roots and, once processed, is sent through the ventral (anterior) roots [6,7]. The spinal nerves, from the spinal cord and through the spinal column, are directed to the trunk and limbs for motor control. From the spinal nerve, motor and sensory innervation of the back is carried out. Taking into account the anatomical structure of the human body, injuries in the PNS can occur at any level, and these can be classified accordingly. Several classifications exist, but in the present report, we have assumed the one presented by Campos and Kimiko [8]. According to these authors, neuropathies are classified as: (a) mononeuropathies (only a single nerve trunk is affected); (b) multiple mononeuropathies (several nerve trunks are affected); and (c) distal polyneuropathies (presenting diffuse and symmetrical involvement of all limbs).

It is now known that fiber sensory neuropathy can occur in patients with various pathologies (diabetes, HIV, sarcoidosis, amyloidosis, etc.). Other diseases such as Guillain–Barré syndrome are associated with demyelinating neuropathies with an immunological component (monoclonal IgM with myelin-associated anti-glycoprotein activity) [9]. Other pathologies resulting from drug intoxication, can also cause neuropathies [10]. Among peripheral nerve lesions, radiculopathy is the most common. These situations have different etiologies, but are mostly due to the compression of the root by canal stenosis. Radiculopathies use to present pain and weakness and can be difficult to differentiate from other mononeuropathies [11].

In the general population, the prevalence of PN is 7–10%, being more frequent in women (8%) than in men (5.7%). Taking into account the age, PN is higher in persons over 50 years (8.9%) compared to persons under 50 years of age with 5.6%. From the etiologic point of view, diabetic neuropathy is the most common cause of chronic PN [11]. In this line, Abbott et al. have observed that one-third of diabetics had painful neuropathic symptoms [12] with an 11% prevalence of PN in diabetic patients [13]. In addition, it has been well established that the occurrence of PN can be associated with a wide variety of diseases and syndromes [14]. In the case of diabetic patients mentioned before, the PNS disorder usually affects the smaller fibers (myelinated β and δ fibers and unmyelinated C-fibres) [14]. Murphy et al. [15] indicate that these somatosensory lesions give rise to adaptation problems in the nervous system, which may be the cause of the presence of spontaneous pain or pain triggered by sensory stimuli.

On the other hand, lumbosacral radiculopathy has a prevalence of 3–5% [16]. However, the incidence of low back pain is 13–31%. Radicular symptoms occur between 12–40% of people with low back pain [16,17]. Many other symptoms are associated with radiculopathy, including paresthesias (63–72%), radiating pain in the lower extremities (35%), and numbness (27%). Muscle weakness has been observed in 37% of radiculopathy patients, 40% of patients have absent ankle reflexes, and 18% absent knee reflexes [16,17].

Polyneuropathy (generalized nerve disease) usually affects the more distal nerves symmetrically [10,11] and is not uncommon during the acute process of Guillain– Barré syndrome. As previously mentioned, other pathologies, including diabetes mellitus, drug toxicity (alcohol abuse), hereditary neuropathies, HIV, inflammatory neuropathies (chronic inflammatory polyradiculoneuropathy), and leprosy, can cause polyneuropathies [18,19]. The prevalence of polyneuropathy seems to be low (1–3%) [19]. However, a study conducted in the United States found that the prevalence of peripheral neuropathy in adults aged 40 years or older was 14.8% [20], indicating that the prevalence of PN increases with age.

However, although multiple consequences of COVID infection have been described, no studies have focused on possible neurological consequences, specifically neuropathies.

### 2.1. Pathophysiology of Neuropathy

PN is characterized by pain, numbness, tingling in the extremities, and slow nerve conduction. Understanding the pathophysiological process of neuropathic pain is complex because it involves different aspects, ranging from the participation of nociceptors, the reaction of the nerve cell membrane, sensitivity to catecholamines, and even the activation of the immune system [21,22]. All these factors are accompanied by inflammatory reactions, vascular alterations, the proliferation of vascular endothelium and vessel smooth muscle, and activation of transcription factors (NF-κB, TGF-β), among others [23,24,25,26,27].

The neuropathy that appears in diabetic individuals is the most studied and best known. Bernard stated that diabetes mellitus was a neurological disease with secondary metabolic manifestations. Contemporary to Bernard, Marchal de Calvi clarified that diabetic neuropathy was a consequence and not the cause of diabetes mellitus [28]. From the metabolic point of view, 3 aspects seem to determine neuropathy: oxidative damage, osmotic unbalance, and inflammation [29]. In diabetic neuropathy, the alterations occur due to the excess glucose outside the cell [30,31], which generates an increase in glucose flux, overactivation of protein kinase C, accumulation of glycosylated products, and increased formation of superoxide radicals.

#### 2.1.1. Inflammatory-Immunological Factors and Neuropathy

Diabetic neuropathy, considered as a predominant inflammatory-autoimmune pathophysiology, is not only limited to lumbosacral radiculopathy, but also includes cervical, thoracic, and cranial neuropathies. These involve axonal loss and leukocyte infiltration. Intraneuronal glucose metabolism is mainly diverted towards the production of proinflammatory and fibrogenic cytokines, as well as inflammation-activated transcription factors including NF-kB (nuclear factor-kB) and TGF-β (transforming growth factor-β) [32,33].

Therefore, making the differential diagnosis of polyradiculopathy implies considering different pathologies, including autoimmune diseases such as inflammatory demyelinating polyradiculoneuropathy, neurosarcoidosis, paraproteinemias, and Sjogren’s disease; viral infections such as cytomegalovirus, HIV, Epstein–Barr virus, and Herpes simplex virus; bacterial infections; protozoal infections such as Toxoplasma; and neoplastic diseases. Several studies [34,35,36] showed a lymphocytic pleocytosis and a high concentration of proteins in the cerebrospinal fluid (CSF). These signs suggest an inflammatory, infectious or neoplastic process [34,35]. Polyradiculopathy tends to improve with symptomatic pain management, rehabilitation, and optimal glycaemic control with intensified insulin therapy [34,35].

#### 2.1.2. Neuropathies Caused by Nutritional Deficiencies and Toxic Agents

Many factors can favor the development of neuropathy such as degenerative diseases, infections, genetic alterations, trauma, ischemia, cancer, inflammation, and immunological diseases, among others. In addition, the relationship between certain neuropathies and deficiencies in specific micronutrients, such as thiamine, niacin, pyridoxine, and cobalamin, is well documented. These vitamins play a key role as coenzymes in metabolic pathways in the nervous system, including energy production, antioxidant protection, biosynthesis of neurotransmitters, and myelin formation. Furthermore, these nutrients work in a synergistic way complementing efficiently their biological actions. Alcohol abuse and malabsorption syndromes are the main causes of vitamin deficiency in modern societies [37]. Regarding toxic agents, poisoning with certain metalloids and metals such as arsenic, golden, lead, mercury, and thallium can result in neuropathy. Contaminated foods and drinks are the main way to reach the organism and cause the disease [2].

## 3. Neuropathy in COVID

According to the literature, patients with SARS-CoV-2 infection (COVID-19) have complained of headache, nausea, vomiting, myalgia, dizziness, hypogeusia, hyposmia, and altered consciousness. Altogether, they are symptoms suggestive of nervous system affectation [38,39]. In this context, Montalvan et al. [40] have reviewed the neurological manifestations of COVID-19 infection and other coronaviruses. However, they have not reviewed chronic or late-onset neurological disease associated with COVID-19 and if the virus can invade the nervous system. The study of other coronaviruses that can infect the nervous system could serve as models to decipher how SARS-CoV-2 affects or invades the nervous tissue [41].

The development of late-onset neurological disease can be explained by genetic factors and host antiviral response [42]. Genetic diversity and rapid evolution are identifiable features of COVID-19 [43]. Therefore, the possibility of late neurological disease associated with COVID-19 infection must be considered, as a result of multiple gene-immune interactions between the virus and the host.

Patients with COVID-19 or who have passed the disease display neuromuscular manifestations such as facial paresis, Guillain-Barré syndrome, symmetrical neuropathy, myopathy, critical illness neuropathy, myalgia, myositis, and rhabdomyolysis [44,45]. In the case of COVID-19, both the central and PNS may be affected. In this context, sensory disturbances are frequent and have a hypoxic and metabolic etiology. The characteristic “cytokine storm” favors severe changes in metabolism and multiple organ failure [46]. “Cytokine storm”, a main mediator of inflammation in COVID-19, is characterized by lymphopenia, increased neutrophil numbers, and high levels of proinflammatory cytokines. Severe neurological complications are the product of SARS-CoV-2 invasion, immune reaction or the hypoxic state generated [47,48]. The post-infectious immune-mediated inflammatory process has been observed in cases of Guillain-Barré syndrome associated with COVID-19.

One of the key systems affected in neurological disorders is the loss of connection between nerve and muscle through the neuromuscular junctions. Defects in signal transmission between nerve endings and muscle membrane are a key feature in many pathologies. We hypothesize that impaired connection at the level of neuromuscular junction could be in part related to the muscular complications observed in COVID patients. In this context, SARS-CoV-2 infected patients present muscle problems (myalgias), indicating a state of muscle injury associated with fatigue [49,50]. In this line, two meta-analysis describe myalgias, muscle pain, and fatigue in 35% of COVID-19 patients [51,52]. Mao et al. [53] reported that 10% of patients with COVID-19 showed muscle lesions associated with an increase in circulating creatine kinase (CK). This clinical picture leads to rhabdomyolysis, characterized by myalgia and fatigue. Several authors [54,55] indicate that in patients with COVID-19, rhabdomyolysis should be suspected if they present localized muscle pain or general weakness [54,55]. This situation seems to be a consequence of muscle invasion by the virus through the membrane-bound angiotensin-converting enzyme-2 (ACE-2) [56]. Moreover, the presence of comorbidities (diabetes and hypertension) potentiates the expression of the ACE-2 in the brain, favoring the neurotropism of the SARS-CoV-2 virus [57].

It should also be taken into account that some drugs applied in the treatment of COVID-19 can cause neurological toxicities and muscle affectations [58]. In this regard, myopathies derived from the use of hydroxychloroquine have been described. This type of myositis is characterized by muscle weakness, but with normal serum CK values [59].

## 4. Management of Neuropathies

The focus of this review will be directed mainly on nutraceuticals. However, a pharmacological approach to neuropathies exists and is directed toward the use of analgesics, anticonvulsant drugs, antidepressants, and topical treatments [60] (Table 1).

Then, we analyzed other elements that have been used in diabetic neuropathies and which seem to have shown positive effects in the treatment of this type of pathology.

### 4.1. α-Lipoic Acid

α-Lipoic acid (ALA), also known as thioctic acid or simply as lipoic acid, is a powerful antioxidant, acting as a coenzyme in mitochondrial reactions in which glucose is converted into energy, such as the Krebs cycle, as well as in the catabolism of α-keto acids or amino acids [61,62,63]. ALA and its reduced form dihydrolipoic acid (DHLA) increase the levels of endogenous antioxidants such as glutathione and coenzyme Q10. In addition, it is able to reduce oxidized vitamins C and E. For this reason, ALA is called the antioxidant of antioxidants or the universal antioxidant. In addition, ALA decreases the production of cytokines and pro-inflammatory factors [61,62,63,64].

Inflammation is an oxidative process, and the modulation and suppression of oxidative damage have been investigated for decades. In Europe, ALA has been extensively studied for the treatment of diabetic neuropathy [65,66,67,68,69]. Treatment with ALA decreases oxidative stress and improves endothelial function in patients with metabolic syndrome and animal models with diabetic neuropathy [70,71]. Both in the short-term (600–1800 mg/day) [65,68] and long-term (4 years), ALA treatments improve neuropathy and muscle weakness with no changes in nerve conduction.

Aside from its antioxidant properties, ALA inhibits the transcription factor NF-kB [72]. NF-kB is found in the main part of cells and initiates the cellular response to stimuli such as stress, inflammation, bacterial or viral antigens, among others [73,74]. ALA inhibits NF-kB-dependent metalloproteinase-9 expression. Furthermore, it decreases in vitro the expression of vascular cell adhesion molecule-1 (VCAM-1) and endothelial adhesion of human monocytes [75,76]. Moreover, ALA prevents the upregulation of intercellular adhesion molecule-1 (ICAM-1) and VCAM-1 in spinal cords and cultured brain endothelial cells [77]. Similarly, ALA inhibits TNF-α activation induced by NF-kB and VCAM-1 expression in the human aorta through a mechanism apparently distinct from its role as an antioxidant [78].

In addition, the downregulation of the CD4 surface area observed in ALA-treated blood mononuclear cells has been proposed to explain in part, the regulation of infiltration of inflammatory cells into the central nervous system [79]. However, the anti-inflammatory properties of ALA require additional studies in humans. Sola et al. [70] observed a significant 15% decrease in serum IL-6 levels after 4 weeks of ALA supplementation (300 mg/day). This finding may be important as IL-6 also modulates the expression of additional inflammatory cytokines, including IL-1 and TNF-α [80,81].

### 4.2. Acetyl-L-Carnitine

Acetyl-L-carnitine (ALC) is an effective dietary supplement for diabetic neuropathy [82,83,84]. ALC is key to a mitochondrial function [85], promoting the expression of nerve growth factors and peripheral nerve regeneration and conduction [86]. Patients with chronic diabetic neuropathy treated with ALC (500–1000 mg 3 times a day) manifested sensitive improvements in visual analog pain scale (VAS) and vibration perception in fingers and toes. In addition, they showed an increase in the number of nerve fibers after 52 weeks of treatment. Moreover, in a randomized study, ALC was shown to reduce pain in antiretroviral (HIV) neuropathy [87].

### 4.3. Vitamin D

Vitamin D deficiency has become a public health problem associated with cardiovascular disease and cancer risk, and autoimmune conditions, and may play a role in the development of diabetes and neurodegenerative diseases. Until the publication of Panfili et al. [88], only one article had addressed the relationship between vitamin D levels and clinical effects in COVID-19 patients [89]. The authors [88] observed that serum vitamin D levels seem to predict the severity of the disease. Nevertheless, studies addressing a connection between vitamin D status and COVID-19-related neuropathy are still scarce [90]. Knowledge regarding other neuropathies could give some keys for future research in neuropathy associated with COVID-19.

In this line, vitamin D deficiency may be considered an independent risk factor in diabetic neuropathy [91]. It appears that one of the etiological causes of diabetic neuropathy is the presence of proinflammatory cytokines. Since vitamin D deficiency upregulates inflammatory mediators (IL-13 and IL-17) in diabetes and diabetic neuropathy, it can be suggested that vitamin D deficiency may be a modifiable risk factor [92,93]. Vitamin D is involved in the coordination of growth, metabolic processes, and immune function, among others [94] (Table 2). In addition, vitamin D increases the expression of two antimicrobial peptides: cathelicidin and β-defensin. Both play a key role in innate immunity [95,96]. The effects of the human cathelicidin LL37 peptide are through interaction with a formyl peptide receptor-like peptide 1 (FPRL1), leading to the recruitment of neutrophils, monocytes, and T cells to infectious loci. Cathelicidin also stimulates apoptosis of infected cells exerting an antiviral effect against a variety of viruses, including HIV and influenza virus [97].

Vitamin D decreases the mediated response of Th1 lymphocytes, particularly in the synthesis of the proinflammatory cytokines IL-2 and interferon-γ (IFN-γ) during secondary or adaptive immunity [98,99]. However, in the case of Th2 lymphocyte inflammation, vitamin D increases cytokine production. Likewise, vitamin D induces the expression of FoxP3 (Forkhead box protein P3), a transcription factor that is directly involved in the function of CD4+ regulatory T cells and in the development and function of Treg lymphocytes. In dendritic cells (DC), vitamin D inhibits differentiation and suppresses the expression of the proinflammatory cytokine IL-12 and the upregulation of the anti-inflammatory cytokine IL-10 [100,101].

The development of Th17 lymphocytes (LTh17) is suppressed by vitamin D. Th17 lymphocytes are involved in the pathogenesis of various autoimmune diseases [102]. In this context, work performed in mice has shown that CD4+ cells with vitamin D receptor deletion showed increased development towards Th17 and increased IL-17 production [102,103]. On the other hand, peripheral mononuclear cells from patients with rheumatoid arthritis and cultured in the presence of vitamin D had reduced levels of IL-17, IL-6, and TNF-α. This observation is indicative of a specific immunosuppressive effect on cytokine production for Th17 development [104]. Based on the presented evidence, it could be hypothesized that vitamin D could be a good adjuvant in the immune response against the SARS-CoV-2 virus and/or act as a modulatory element. Future research needs to be directed in this line.

### 4.4. B Vitamins

In the context of the nervous system, certain vitamins of group B (B1, B6, and B12) play a key role in axonal transport, neuronal excitability, and neurotransmitter synthesis. In addition, these vitamins exert a main function in some painful conditions such as neuritis, lumbago, and sciatica. Supplements of these vitamins have been used in the treatment of lumbar vertebral syndromes, chronic headaches, trigeminal neuralgia, and chronic pain related to diabetic polyneuropathy and rheumatoid arthritis [105]. Vitamins B1 (thiamine), B6 (pyridoxine), and B12 (cyanocobalamin) can significantly reduce the severity and duration of neuropathic pain.

It has been described even in general biochemistry textbooks [106] that vitamin B deficiency states have an adverse effect on the nervous system. Vitamin B1 is involved in the transmission of nerve impulses, and B6 is involved in the metabolic transformation of amino acids. Pyridoxal phosphate (the active form of vitamin B6) is necessary for the synthesis of sphingolipids in the formation of myelin. B12 (cobalamin) is converted to methylcobalamin and 5-deoxyadenosylcobalamin, participating in cell growth and replication [107]. Common features of B1, B2, and B12 deficiency include peripheral neuropathy, lack of motor synchronization, mental confusion, depression, and malaise.

The general effects of B1, B6, and B12 vitamins on axonal conduction may contribute to immediate analgesia through different mechanisms [108]:-Vitamin B1 may exert an important biophysiological role in nerve conduction, and excitability;-Vitamin B12 may selectively block sensitive nerve conduction;-Prolonged pain inhibition may be due to potential vitamin interactions with intraspinal and supraspinal receptors in different systems, via tonically released endogenous opioids or inhibitory non-opioid neurotransmitters, such as serotonergic and γ-aminobutyric acid (GABA).

Separated administration of the above-mentioned B vitamins displays an anti-pain effect. However, none of them, in isolation, is as effective as in combination. The effect of the combination of the different B vitamins is important because, in different models of neuropathic pain, vitamin B12 can enhance the efficacy of vitamins B1 and B6 [108,109,110,111]. Several clinical trials support the use of these B vitamins in diabetic neuropathy despite the absence of evidence of deficiency states [109,110,111]. In diabetic neuropathy, the vitamin B complex (B1, B6, and B12) has been shown to increase nerve fiber density and increase 2-point discrimination [109,110]. Both vitamin B6 and B12 are cofactors in the impairment of peripheral nerve function in diabetic subjects. Vitamin B12 has been found to facilitate myelinogenesis and nerve regeneration [111].

#### 4.4.1. Vitamin B1 (Benfotiamine)

Benfotiamine is a soluble derivative of vitamin B1 with high bioavailability, and is involved in the control of oxidative stress by reducing markers of endothelial dysfunction in type 2 diabetic subjects [112]. Symptomatic improvement of pain after treatment of diabetic neuropathy with a dose of 600 mg/day is observed after 6 weeks [113,114].

Three enzyme systems are central in thiamine-dependent glucose metabolism in the brain: pyruvate and α-ketoglutarate dehydrogenase complexes and cytoplasmic transketolase. These enzymes use thiamine pyrophosphate as a coenzyme, representing 80% of the total thiamine in nervous tissue [115]. Moreover, oxidative stress, which damages different cellular structures, can be mitigated by the administration of thiamine, acting as a scavenger of free radicals and reactive oxygen species (ROS). Thiamine deficiency, which leads to increased ROS, initiates cell membrane damage, affecting neuronal ion channels and changes in transporters and microglia.

#### 4.4.2. Vitamin B6 (Pyridoxal-5′-Phosphate)

First of all, it should be noted that high and sustained doses of vitamin B6 can cause severe sensory ataxic neuropathy. In this regard, doses of 200 mg/day, and higher, appear to generate toxicity [116]. Both vitamin B6 deficiency and overdose can lead to neurological complications, including neuropathy. Adequate and safe doses to treat deficiency states should be in the range of 50–100 mg/day [117]. It should also be noted that low circulating levels of vitamin B6 are associated with increased C-reactive protein (CRP) [118,119]. Moreover, high vitamin B6 intake is associated with protection against inflammation [120]. Therefore, and taking into account that vitamin B6 is a coenzyme for more than 100 metabolic reactions, it may be of dietary interest to include this supplement as an anti-inflammatory and cytokine production regulator. In this line, oral benfotiamine administration for eight weeks seems to exert an improvement on the vibration detection threshold. However, high oral doses of thiamine and pyridoxines appear to have a greater short-term effect on neuropathic symptoms than low doses [121].

#### 4.4.3. Vitamin B12 (Methylcobalamin)

Vitamin B12 deficiency is relatively common, especially in older people when intestinal absorption decreases. Cobalamin and its derivative (methylcobalamin) are necessary for the maintenance of the peripheral nervous system. Cobalamin participates in the conversion of homocysteine to methionine, which is involved in the biosynthesis of lecithin, necessary for myelination and nerve regeneration. Methylcobalamin is implicated in nerve damage following peripheral nerve injury in animal models [122]. Methylcobalamin (500 mg, 3 times/day) has been found to improve peripheral symptoms of diabetic neuropathy, but no change in motor and sensory nerve conduction was observed after 4 months [111]. In this context, it has been reported that the combination of methylcobalamin, ALA, and pregabalin provides pain relief, and improves sleep interferences and nerve function [123].

## 5. Conclusions

Neuropathy has multiple etiologies, with being infectious being one of them. One of these current situations is COVID-19 infection, the manifestations of which are very varied, affecting different organs and systems. However, the effect of COVID involvement is not well understood, due to few studies published at present.

In this review, we have analyzed the factors involved in the development of neuropathy, without going into the details of symptomatology and diagnosis. We have tried to clarify the possible mechanisms by which the pathology generated by COVID-19 can lead to the development of neuropathies. Based on this and the knowledge coming from studies on other types of neuropathies, we have proposed a nutraceutical treatment.

In general, neuropathies are debilitating and painful diseases sometimes refractory to pharmacological treatments. For this reason, we have proposed alternative therapeutic approaches, including nutraceuticals that individually, or in combination, have been shown to be effective in relieving painful symptoms. These could be new lines of research in the future for alternative treatments in COVID-19 affected patients.

## Figures and Tables

**Table 1 biomedicines-10-01051-t001:** Proposed pharmacological treatments for neuropathies based on GRADE (Grading of Recommendations, Assessment, Development, and Evaluation) system.

1st Line (Strong Recommendation)	2nd Line (Moderate Recommendation)	3rd Line (Weak Recommendation)
GABA analogs	Alkaloids	Strong opioids
(gabapentin, pregabalin)	(capsaicin patches)	(morphine, oxycodone,
Inhibitors of serotonin and noradrelanine reuptake	Local anesthetic	tapentadol)
	(lidocaine patches)	Neurotoxins
(duloxetine, venlafaxine)	Opioid analgesics	(botulinum toxin A (BTX-A))
Tricyclic antidepressants	(tramadol)	
(amitriptyline, desipramine,		
clomipramine, imipramine,		
nortriptyline)		

**Table 2 biomedicines-10-01051-t002:** Effects of vitamin D in the immune system.

Decrease in lymphocyte activation (Th1).
Decrease in IL-12 production
Decrease in IFN-γ production
Decrease in Th17 through decreased production in IL-6 and IL-23
Tolerogenic dendritic cells: Th2, Treg, IL-10
Increase in lymphocytes Treg CD25+/Foxp3+
Decrease in the proliferation of T lymphocytes through decreased production in IL-2
Decrease in proliferation and differentiation of B cells (inhibition of differentiation in plasma cells and memory B cell)
Decrease in synthesis of immunoglobulins
Inhibition of NFkB (via p105/p50) in B naive cells

Abbreviations used: FoxP3 gene, Forkhead box protein P3; NFkB, nuclear factor-kB; Treg, regulatory T cells.

## Data Availability

Not applicated.

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
