# Peer review of "Peripheral Neuropathies Derived from COVID-19: New Perspectives for Treatment"

_biomedicines, 2022, doi:10.3390/biomedicines10051051_

Round 1

Reviewer 1 Report

The issue raised by the Authors is undoubtedly worth exploring. However, I have important critical comments about the article:

  1. In section: „3. Neuropathy in COVID:, the authors write: "In this regard, the neurological transmission of hepatitis virus (HV) has been studied in mice. It has been reported that HV is transported to 152 the spinal cord through well-defined neuroanatomical pathways [40]". It is absolutely unclear how the authors of this review want to link the above-mentioned HV information to the Sars-CoV2 virus. This requires clarification.
  2. In the same section, between lines 172 and 185, the authors devote a great deal of space to the muscle damage induced by COVID-19, without explaining how they want to relate this data to neuropathy, which is necessary.
  3. The entire chapter on "Neuropathy in COVID" is unsatisfactory - and it is very important from the point of view of the whole article. The authors should devote more space and attention to the data on the relationship between COVID and the neuropath, and when too little data is available, they should be tempted to use their own stories and concepts. Moreover, the authors provide no data on the prevalence of neuropathy in COVID-19 or at least on the prevalence of non-specific symptoms that may suggest neuropathy.
  4. Although the purpose of the article was to discuss the issues related to COVID-19-induced neuropathy, the article is simply devoted to neuropathy and by the way, the aspect related to COVID-19 was very sparingly and insufficiently discussed.

Author Response

REVIEWER-1

The issue raised by the Authors is undoubtedly worth exploring. However, I have important critical comments about the article:

  1. In section: “3. Neuropathy in COVID”: the authors write: "In this regard, the neurological transmission of hepatitis virus (HV) has been studied in mice. It has been reported that HV is transported to the spinal cord through well-defined neuroanatomical pathways [40]". It is absolutely unclear how the authors of this review want to link the above-mentioned HV information to the Sars-CoV2 virus. This requires clarification.

ANSWER: We indicate in this section that it has been described nervous system affectations caused by SARS-CoV-2 infection. However, the way by which the virus enters in the nervous system is not clear yet. We have emphasized the idea to develop working hypothesis taking as models other viruses that are capable to infect the nervous system. Likely, the selection of the hepatitis virus could be an example but others can inspire readers for future research. We have changed the sentence giving a more general perspective (see lanes 164-166).

  1. In the same section, between lines 172 and 185, the authors devote a great deal of space to the muscle damage induced by COVID-19, without explaining how they want to relate this data to neuropathy, which is necessary.

ANSWER: One of the key systems affected in neurological disorders is the loss of connection between nerve and muscle through the neuromuscular junctions. Defects in signal transmission between nerve endings and muscle membrane is a key feature in aging or pathologies such as amyotrophic lateral sclerosis. We hypothesize that impaired connection at the level of neuromuscular junction could be in part related to the muscular problems observed in COVID patients. We have clarified this point in the text (see lanes 184-188).

  1. The entire chapter on "Neuropathy in COVID" is unsatisfactory - and it is very important from the point of view of the whole article. The authors should devote more space and attention to the data on the relationship between COVID and the neuropathy, and when too little data is available, they should be tempted to use their own stories and concepts. Moreover, the authors provide no data on the prevalence of neuropathy in COVID-19 or at least on the prevalence of non-specific symptoms that may suggest neuropathy.

ANSWER: We believe that the changes we have performed in this section have focused the role of SARS-CoV-2 infection and the connection to neuropathy. It is true that more research is necessary and for this reason some features are presented as hypothesis to test in future research.

  1. Although the purpose of the article was to discuss the issues related to COVID-19-induced neuropathy, the article is simply devoted to neuropathy and by the way, the aspect related to COVID-19 was very sparingly and insufficiently discussed.

ANSWER: The purpose of the review is not to discuss the issues related to neuropathy induced by COVID-19. The review focuses on the application of nutraceutical strategies in order to improve the symptomatology of neuropathy and neuropathic pain in patients that suffered from COVID-19. We aim is to provide an alternative vision to traditional medical-pharmacological treatment through nutraceuticals. This goal was clearly indicated in the ABSTRACT (lanes 25-28).

Reviewer 2 Report

This manuscript is a review of peripheral neuropathy caused by COVID-19 infection and potential roles for vitamins/supplements in treatment thereof. The subject is interesting and timely. The manuscript would be improved by a few clarifications, most of which are quite minor.

Specific comments:

- Section 2, line 59-61: I am not sure of the meaning of this sentence. I believe the authors might be describing sensory and motor spinal roots; if this is the case, the information about anterior and posterior roots is reversed – efferent motor nerves are in the anterior (ventral) roots, and sensory information is received through the dorsal (posterior) roots.

- It might be useful to have a subsection in section 2 to discuss peripheral neuropathies caused by nutritional deficiencies/toxicities. There is some information about this in the management section, but given the title and introduction it would make sense to tell a little about this here as well.

- Line 151-152: The authors should specify that this is Mouse Hepatitis Virus specifically. This information would be helpful for the readers, since MHV is also a coronavirus. Knowing this helps us understand why MHV is relevant to COVID-19 related peripheral neuropathy.

- Section 4, line 191: It is no longer accurate that there are no specific treatments proposed for COVID19.

- Table 1: Where did these recommendations come from? Is this the recommendation from the authors, or is there a reference that should be here to support these recommendations?

- Table 1: It is not clear what is the meaning of the end of the table containing “Weak recommendation: Cannabinoids vs Valproate; Strong recommendation: Levitiracetam vs Mexiletine.”

- Line 250: Should this sentence read “…vitamin D deficiency upregulates inflammatory…”?

- The vitamin D section should draw a more direct connection between vitamin D status and COVID-19 related neuropathy. As it stands the authors make a case for Vitamin D in COVID symptoms, but not specifically for COVID neuropathy.

- Lines 341-344: These sentences appear to be out of place. Should they be in the thiamine (B1) section?

- Conclusions (section 5): What are the authors’ recommendations based on the information reviewed here?

Minor points:

- Line 36: the word “revision” should be “review” or “systematic review” I believe.

- Section 4.4 headings: Usually this would be rendered as simply “B Vitamins” in English.

Author Response

REVIEWER-2

This manuscript is a review of peripheral neuropathy caused by COVID-19 infection and potential roles for vitamins/supplements in treatment thereof. The subject is interesting and timely. The manuscript would be improved by a few clarifications, most of which are quite minor.

Specific comments:

- Section 2, line 59-61: I am not sure of the meaning of this sentence. I believe the authors might be describing sensory and motor spinal roots; if this is the case, the information about anterior and posterior roots is reversed – efferent motor nerves are in the anterior (ventral) roots, and sensory information is received through the dorsal (posterior) roots.

ANSWER: Mistake has been corrected accordingly (see lanes 60-61).

- It might be useful to have a subsection in section 2 to discuss peripheral neuropathies caused by nutritional deficiencies/toxicities. There is some information about this in the management section, but given the title and introduction it would make sense to tell a little about this here as well.

ANSWER: We have included a new section (Section 2.1.2) to show briefly this information (see lanes 144-156).

- Line 151-152: The authors should specify that this is Mouse Hepatitis Virus specifically. This information would be helpful for the readers, since MHV is also a coronavirus. Knowing this helps us understand why MHV is relevant to COVID-19 related peripheral neuropathy.

ANSWER: We have changed the sentence according to Reviewer-1 suggestions and give a more general perspective. The idea is to motivate research among readers to decipher how SARS-CoV-2 can affect the nervous system and cause neuropathy (see lanes 164-166).

- Section 4, line 191: It is no longer accurate that there are no specific treatments proposed for COVID19.

ANSWER: The sentence has been changes accordingly (see lanes 206-207).

- Table 1: Where did these recommendations come from? Is this the recommendation from the authors, or is there a reference that should be here to support these recommendations?

ANSWER: The pharmacological treatments proposed in Table 1 are based in the GRADE (Grading of Recommendations, Assessment, Development and Evaluation) system. This system evaluates the quality of treatments based in the evidence, regarding tolerability and safety. In this line, GRADE proposes a first line of treatment (strong recommendation), a second line (moderate recommendation) and a third line (weak recommendation). See new reference 60 indicated in the new version of Table 1.

- Table 1: It is not clear what is the meaning of the end of the table containing “Weak recommendation: Cannabinoids vs Valproate; Strong recommendation: Levitiracetam vs Mexiletine.”

ANSWER: Table 1 has been changed accordingly.

- Line 250: Should this sentence read “…vitamin D deficiency upregulates inflammatory…”?

ANSWER: Sentence has been corrected (see lane 268).

- The vitamin D section should draw a more direct connection between vitamin D status and COVID-19 related neuropathy. As it stands the authors make a case for Vitamin D in COVID symptoms, but not specifically for COVID neuropathy.

ANSWER: Studies addressing a connexion between vitamin D status and COVID-19 related neuropathy are still scarce. We present the evidence accumulated in other neuropathies and proposed new lines of research (see lanes 262-265).

- Lines 341-344: These sentences appear to be out of place. Should they be in the thiamine (B1) section?

ANSWER: The sentence proposes a treatment with high oral doses of thiamine and pyridoxine. In the previous section (4.4.1), we presented the information regarding benfotiamine. In next section (4.4.2), the information regarding pyridoxal is presented. The last sentence wants to reflect a final conclusion of the pharmacological use of both vitamins. We guess that the sentence is well placed there.

- Conclusions (section 5): What are the authors’ recommendations based on the information reviewed here?

ANSWER: Since the scarce information regarding COVID-19 related neuropathy, we only can propose new lines of research in this topic. To give recommendations is at present too risky because we need to accumulate more knowledge. We hope that this review will inspire readers

in this context. See lanes 390-391.

Minor points:

- Line 36: the word “revision” should be “review” or “systematic review” I believe.

ANSWER: The word has been corrected (see lane 36).

- Section 4.4 headings: Usually this would be rendered as simply “B Vitamins” in English.

ANSWER: The heading of Section 4.4 has been changed accordingly (see lane 302).

Round 2

Reviewer 1 Report

The article as it stands is satisfactory. The changes made by the authors following the reviewers' comments are sufficient.